# Perilipin 5 and Lipocalin 2 Expression in Hepatocellular Carcinoma

**DOI:** 10.3390/cancers11030385

**Published:** 2019-03-19

**Authors:** Anastasia Asimakopoulou, Mihael Vucur, Tom Luedde, Silvia Schneiders, Stavroula Kalampoka, Thomas S. Weiss, Ralf Weiskirchen

**Affiliations:** 1Institute of Molecular Pathobiochemistry, Experimental Gene Therapy and Clinical Chemistry (IFMPEGKC), RWTH University Hospital Aachen, 52074 Aachen, Germany; aasimakopoulou@ukaachen.de (A.A.); silschneiders@ukaachen.de (S.S.); kalabokastivi@yahoo.gr (S.K.); 2Department of Internal Medicine III, RWTH University Hospital Aachen, 52074 Aachen, Germany; mvucur@ukaachen.de (M.V.); tluedde@ukaachen.de (T.L.); 3Children’s University Hospital (KUNO), Center for Liver Cell Research, University Hospital Regensburg, 93053 Regensburg, Germany; Thomas.Weiss@klinik.uni-regensburg.de

**Keywords:** HCC, LCN2, PLIN5, AFP, cancer, liver

## Abstract

Hepatocellular carcinoma (HCC) is one of the most prevalent and deadly cancers worldwide. Therefore, current global research focuses on molecular tools for early diagnosis of HCC, which can lead to effective treatment at an early stage. Perilipin 5 (PLIN5) has been studied as one of the main proteins of the perilipin family, whose role is to maintain lipid homeostasis by inhibiting lipolysis. In this study, we show for the first time that PLIN5 is strongly expressed in tumors of human patients with HCC as well as in mouse livers, in which HCC was genetically or experimentally induced by treatment with the genotoxic agent diethylnitrosamine. Moreover, the secreted acute phase glycoprotein Lipocalin 2 (LCN2) established as a biomarker of acute kidney injury, is also proven to indicate liver injury with upregulated expression in numerous cases of hepatic damage, including steatohepatitis. LCN2 has been studied in various cancers, and it has been assigned roles in multiple cellular processes such as the suppression of the invasion of HCC cells and their metastatic abilities. The presence of this protein in blood and urine, in combination with the presence of α-Fetoprotein (AFP), is hypothesized to serve as a biomarker of early stages of HCC. In the current study, we show in humans and mice that LCN2 is secreted into the serum from liver cancer tissue. We also show that AFP-positive hepatocytes represent the main source for the massive expression of LCN2 in tumoral tissue. Thus, the strong presence of PLIN5 and LCN2 in HCC and understanding their roles could establish them as markers for diagnosis or as treatment targets against HCC.

## 1. Introduction

PLIN5 belongs to the family of perilipins consisting of 5 distinct members (PLIN1-5), which are major structural proteins located on the surface of lipid droplets known to serve as factors for lipid homeostasis by modulating lipid storage, with crucial roles in diseases characterized with lipid manifestations [1,2]. PLIN5 is the latest discovered member of this protein family expressed specifically in fatty acid oxidizing tissues such as heart, liver, adipose tissue and muscle. PLIN5 is encoded by nine exons and is located on chromosome 17 in mice and chromosome 19 in human beings [3,4]. PLIN5 mediates mitochondrial function, regulating fatty acid storage and release to maintain lipid droplet homeostasis [5,6]. Most of the studies performed on PLIN5 have focused on its function in the heart due to its high oxidative capacity. In the liver, a primary study pointed to a protective role of PLIN5 against hepatic lipotoxicity [7].

A very recent study localized the expression of PLIN5 in numerous healthy and diseased human tissues, including liver obtained from subjects with steatosis, acute liver injury as well as mitochondrial deficiency syndrome [4]. Recently, we showed that hepatic PLIN5 expression is regulated by LCN2, controlling intracellular lipid droplet formation in experimentally-induced steatosis [8].

Related to cancer, limited studies analyzed perilipins. Tumor cells are generally characterized by an unstoppable proliferation, which requires high-energy fuels leading to a reprogrammed cellular metabolism [9]. The differentiation of malignant cellular metabolism is where members of the perilipin family could interfere. However, the role of perilipins in cancer has not been yet widely studied. PLIN1 has been shown to be differentially expressed in breast cancer subtypes, while it has proven to serve as a marker to differentiate between liposarcoma and adipocytic sarcomas [10,11]. Furthermore, it has also been proved to be co-expressed with the PLIN2 and PLIN3 in HCC [12]. PLIN2 has been suggested to act as a prognostic marker in clear cell, renal cell carcinoma, where it weakens cell invasion and migration pathways [13]. PLIN2 and PLIN3 have been shown to be higher expressed in tissues of pancreatic cancer, proving the importance of lipid metabolism in the tumor microenvironment [14]. PLIN1-3 have been assigned important roles in tumorigenesis that still need further investigation to decide their practicality in cancer diagnosis and therapy [15]. Another study has recently reported the presence of PLIN5 in cell renal carcinoma, liposarcoma and rhabdomyosarcoma [4]. However, no studies have yet mentioned the presence of PLIN5 in liver cancer. In the current study, we show for the first time that PLIN5 is highly expressed specifically in tumoral areas of HCC livers in human patients as well as in experimental HCC mouse models.

Moreover, LCN2 is a 25-kDa deeply studied secreted protein that belongs to the family of lipocalins, which are characterized by their ability to transport small lipophilic ligands. LCN2 has recently attracted the attention to its role in prognosis and diagnosis in various cancers, including colorectal and breast cancer [16,17]. We recently reviewed the potential roles of LCN2 in diagnosis and prognosis in prostate, pulmonary and hepatic cancer [18]. We came to the conclusion that LCN2 has widely heterogeneous roles in different cancer types, affecting frequently in a contradictory manner the cell proliferation, migration, invasion and metastasis pathways, leaving the presence of LCN2 in the tumor environment under the need of vast investigation [18].

Specifically in HCC, LCN2 expression has been described in several in vivo and in vitro studies, where it is always higher expressed in the tissue as well as in the serum of HCC patients compared to healthy subjects [18]. LCN2 expression inhibits proliferation, invasion and metastasis, with the ability to reverse the epithelial to mesenchymal transition process, suggesting LCN2 as a potential metastasis suppressor and a therapeutic target against HCC [19]. In the same study, LCN2 was shown to be localized in tumoral areas. Furthermore, LCN2 overexpression induced apoptosis of HCC cells through mitochondrial activity in HCC cell lines, reporting LCN2 once again as a potential therapeutic target for the disease [20]. In line, blood LCN2 levels have very recently been assigned the quality of the prognosis marker of the survival in chronic liver diseases complicated by HCC [21].

We have previously studied the role of LCN2 in hepatic injury models in vivo and in vitro, showing that hepatocytes are the major source of LCN2 expression during hepatic damage [22,23,24]. In the present study, we show that LCN2 is strongly expressed in human HCC tissue, as well as in the livers of different murine HCC models with pathogenically distinct trigger (inflammatory or genotoxic) [25,26,27]. The major sources of LCN2 expression in HCC are AFP-positive hepatocytes. Secondly, Myeloperoxidase (MPO) expressing inflammatory cells as well as Cluster of Differentiation 90 (CD90)-positive cells showed also strong expression of LCN2. Under normal conditions, AFP is a major plasma protein produced by the liver during fetal life, and its presence in serum or urine in adults is normally very low, whereas its elevation is correlated to hepatic cancer. AFP has been the most widely used serum biomarker for HCC diagnosis as well as a screening indicator of higher HCC risk patients [28,29,30,31]. The tight correlation of AFP with HCC and its involvement within key cellular functions, such as cell growth, differentiation, apoptosis, angiogenesis, and immune regulation has made AFP also a target for liver cancer immunotherapy [32,33,34].

The AFP immune-expression has been described to be restricted to HCC within their neoplastic hepatocyte and infiltrated basophilic cells while non-tumoral liver and hyperplastic nodules are AFP negative [35]. The data presented in our study link LCN2 and AFP in the pathogenesis of HCC.

## 2. Results and Discussion

Up to date, only a few studies have revealed the participation of perilipin proteins in tumor environments [15,36,37,38,39]. The perilipin family has been assigned limited presence or functions in HCC [12,15]. However, there is not so far a single report connecting the most recent discovered family member of the perilipin family, i.e., PLIN5, with HCC. Therefore, we here tested for its expression in human HCC biopsies as well as in livers of different murine HCC models with pathogenically distinct trigger (inflammatory or genotoxic) [25,26,27]. Initially, testing via RT-qPCR revealed that, while the transcriptional levels of the *Plin2*, *Plin3* and *Plin4* genes in mice only tended to increase in non-tumoral and tumoral regions of HCC livers, *Plin5* was strongly and significantly upregulated in the HCC tumoral area, indicating a possible direct effect of *Plin5* transcription to the tumor development (Figure 1A).

In HCC, PLIN2 and PLIN3 have been reported earlier to be upregulated, inducing the lipogenic pathway as HCC is characterized by more PLIN2- and PLIN3-positive lipid droplets [12,40,41]. Western blot analysis revealed that the PLIN2, PLIN3 and PLIN4 proteins were indeed induced in mouse HCC livers, while the expression of PLIN5, a protein with normally low expression in the liver, was mainly induced in the tumoral areas of the HCC livers (Figure 1B). This finding was confirmed by immunohistochemistry, both in mouse and human HCC biopsies, revealing that indeed PLIN5 is highly expressed in HCC livers with an overwhelming local expression in hepatoma nodule formations found in HCC livers (Figure 2).

The mechanistic functions of other perilipins in HCC has been up to now assigned to pathways related to cell proliferation, early tumorigenesis and lipogenesis regulation in HCC [15]. PLIN5 is a protein that until now has only been studied in mechanisms related to triglyceride metabolism, lipid droplet homeostasis and mitochondrial function in liver tissue in conditions other than cancer [5,6,7,8]. Our finding of its overexpression in developed tumors in HCC paves the way for analyzing the possible functions and practical usability of PLIN5 from diagnosis to treatment.

On the other hand, LCN2, an iron-binding glycoprotein, has been previously shown to be preferentially expressed in HCC with functions in proliferation, invasion, metastasis and apoptosis [19,20,42]. Serum from healthy subjects and also HCC patients was analyzed by Western blot to reveal that LCN2 has increased secretion in the serum of HCC patients in comparison to healthy subjects (Figure 3A,B). Since serum LCN2 increases drastically in kidney damage, serum from patients with kidney injury was used as a positive control [43]. AFP as a secreted protein was also tested in the same samples, as its presence has been used the last years in HCC diagnosis [28,30,31]. As expected, AFP expression was found elevated in the serum of HCC patients. α_2_-Macroglobulin (α_2_M) expression and Ponceau S staining of the Western blot membrane were used as loading controls. Moreover, blood smear analysis for LCN2 confirmed the presence of more LCN2 positive cells in the blood circulation of HCC patients (Figure 3C).

In our mouse models of HCC and human biopsies, we observed a strongly elevated expression of LCN2 in human liver, however not in a uniform pattern. LCN2 could be observed in tumor areas, adjacent tissue, and necrotic areas, as well as inflammatory infiltrates (Figure 4 and Figure 5).

To determine the source of LCN2 in HCC liver, we proceeded with double immunofluorescent staining of LCN2 with AFP, a biomarker of damaged tumoral hepatocytes [35] and MPO, representing a marker of infiltrating neutrophil granulocytes and monocytes [44]. Both staining combinations showed in mouse and human identical localization. All cells positive for either AFP or MPO were also strongly expressing LCN2 (Figure 6 and Figure 7).

AFP has been studied intensely during the last decades. Up to date, AFP remains one of the most useful tumor markers in screening for HCC [45]. While healthy adults have repressed AFP synthesis and concentration in their blood, more than 70% of HCC patients have high concentrations of AFP in their serum. AFP has been shown to be a specific tool for diagnosis and for differentiating HCC from non-malignant hepatopathy [46,47,48]. Moreover, its usefulness as a prognostic marker has also been thoroughly studied. Concentrations higher than 400 ng/mL have been linked to low survival rates, tumor size, and tumor differentiation [49,50,51,52]. A recent review also described the growing impact of AFP as a prognostic marker to predict liver transplantation success [53]. The molecular functions of AFP in HCC have been associated with oncogenic effects including cell proliferation induction by regulation of cell cycles [19,20,54]. AFP has also been reported to stimulate invasiveness and metastasis in hepatoma cell lines and mouse HCC models [55]. Similarly, LCN2 is another useful secreted serum protein used diagnostically as a marker of organ injury. Upregulation of LCN2 and its receptor LCN2R (i.e., the solute carrier family 22 member 17, SLC22A17) in HCC, have been reported to serve as prognostic markers for overall survival [56]. LCN2 has also been shown to induce apoptosis via mitochondrial pathways in HCC [52]. Moreover, LCN2 has been reported to negatively modulate epithelial-to-mesenchymal transition in HCC cells, suggesting that LCN2 could be studied further as a metastasis suppressor or a treatment target [57].

Since AFP is a tool with pleiotropic qualities related to hepatic cancer, the expression of LCN2 in AFP positive cells in HCC could give LCN2 properties in oncogenic-related processes, as well as diagnosis and prognosis. However, further investigations are necessary to understand the precise linkage of AFP and LCN2.

The co-localization of MPO and LCN2 was relatively expected, as LCN2 was originally isolated from human neutrophils [58]. Furthermore, we recently reported the co-expression of LCN2 in MPO positive cells in a model of Lipopolysaccharide-induced inflammatory liver injury in mice [23].

The increased LCN2 expression from MPO positive cells in HCC could indicate a direct pro-inflammatory defensive function against causes of infiltrate recruitment during HCC progression, which confirms other reports indicating that LCN2 could induce facilitation of inflammatory cells recruitment as an intrinsic sensor of defense upon injury [23,59,60].

Having realized that the injured cancerous hepatocytes and inflammatory cells are the major sources of LCN2 in HCC livers, we proceeded with double staining of LCN2 with CD90. CD90 is a glycoprotein which is highly expressed in bone marrow derived stem cells and hepatic progenitor cells [61,62,63,64]. Although adult and fetal hepatic progenitor cells express CD90, adult hepatocytes do not [65]. CD90 has been suggested as a biomarker in several tumors, since it participates in processes including apoptosis, adhesion, migration, fibrosis and cancer. Specifically in HCC the CD90 expression analysis has been reported to increase drastically in transcriptional and protein level in tumors compared to both neighboring cirrhotic tissue and normal liver [65,66].

Thus we wanted to see whether hepatic progenitor cells rather than hepatocytes and inflammatory cells express LCN2 in HCC, we found that in liver sections of HCC mice but not in human HCC, LCN2 is strongly expressed by CD90 positive cells, which are increased in numbers in HCC livers compared to non-HCC biopsies (Figure 8). The co-expression of CD90 and LCN2 could indicate a species dependent mechanism of LCN2 in the process of cell differentiation during HCC progress.

We confirmed the LCN2 overexpression in HCC livers in mouse liver, both in transcriptional and protein levels specifically in the tumoral area extracts, as well as the parallel expression of AFP, MPO and CD90 (Figure 9). LCN2 transcription and protein expression seem to be mostly aroused in HCC tumoral areas, indicating that LCN2 needs to be further investigated to reveal specific mechanistic roles in HCC that could give later answers in understanding development and progress of HCC.

## 3. Materials and Methods

### 3.1. Human Samples

Experimental procedures were performed according to the guidelines of the charitable state controlled foundation HTCR (Human Tissue and Cell Research, Regensburg, Germany), with the patient’s written and informed consent in the German language. More details about the charitable state-controlled foundation HTCR are given elsewhere [67,68]. All analysis involving human tissues and cells have been carried out in accordance to The Code of Ethics of the World Medical Association (Declaration of Helsinki). A total of 36 human samples, including 17 samples of human HCC patients, 10 non-HCC, as well as serum from 5 healthy patients and 4 patients with kidney injury were analyzed. Samples were obtained from patients undergoing liver segmental resections or liver hemi-hepatectomies. All HCC tissues included tumoral and fibrotic regions. HCC samples were obtained from six female and eleven male patients aged between 33–85 years. Except for one female and one male positive for Hepatitis C (HCV), all other subjects were Hepatitis B (HBV) and HCV negative, while the serum AFP ranged between 1.5 µg/L and 379.2 µg/L, respectively. Patients’ characteristics including data on liver damage serum markers, viral infections and liver scoring of all samples according to the TNM system, are reported in Appendix A. The HCC subjects had an occasional to abusive consumption of alcohol.

### 3.2. Human Blood Collection

After collection of the whole blood, it was stored at room temperature (RT) for 30 min to allow clotting. The serum was separated from the clot by centrifugation at 5000× *g* for 10 min in a refrigerated centrifuge.

### 3.3. Animal Experiment

HCC samples from genetically modified mice were obtained from different genetic models (TRAF2/RIPK1^LPC-KO^, NEMO^LPC-KO^, TAK1^LPC-KO^) [25,26,27] with spontaneous pathogenically distinct inflammatory HCC development, or from wild type mice treated with the genotoxic agent (diethylnitrosamine (DEN)), that were obtained by treatment following a standard protocol [69]. All animals from which samples used in this study were treated in full compliance with the guidelines for animal care and the protocols used were approved by the institutional German Animal Care Committee (LANUV, Recklinghausen, Germany; Az.: 84-02.04.2011.A173; permission date: 24 August 2011).

### 3.4. Protein Analysis

Total protein from human or mouse liver tissue, that had been snap-frozen in liquid nitrogen, was prepared as described before [8]. Equal amounts of total protein (100 μg/lane) or serum (5 μL for mice and 1 μL for human) were mixed with NuPAGE™ LDS electrophoresis sample buffer (Invitrogen, Thermo Fisher Scientific, Darmstadt, Germany) implemented with dithiothreitol (DTT) as a reducing agent. Denaturation of protein samples, electroblotting and Western blot were applied as described previously [8]. The primary antibodies were diluted in 2.5% (*w/v*) non-fat milk powder in Tris-buffered saline with Tween 20 (TBST) prior of membrane incubation. The primary antibodies used in this study (Table 1) were visualized with anti-mouse, anti-rabbit or anti-goat IgG secondary antibodies (all from Santa Cruz Biotech, Santa Cruz, CA, USA) with the SuperSignal chemiluminescent substrate (Pierce, Bonn, Germany). α_2_M was used as a loading control for serum samples as a relatively stable plasma protein [70]. Western blot results were densitometrically analysed using the publicly available ImageJ software (version 1.52a, https://imagej.nih.gov/ij/download.html).

### 3.5. RNA Expression

Total RNA from mouse liver tissue, that had been snap-frozen in liquid nitrogen, was extracted and purified with DNAse digestion as described before [24]. Complementary DNA (cDNA) synthesis followed from reverse transcription of 1 μg of purified RNA in a final volume of 25 μL using Superscript II reverse transcriptase and random hexamer primers (all reagents from Invitrogen). The synthetized cDNA was amplified in a 25 μL volume using SYBR GreenTM qPCR SuperMix (Applied Biosystems, Life Technologies, Darmstadt, Germany) via quantitative real-time polymerase chain reaction (RT-qPCR), as recently described [24]. The primer pairs used for the RT-qPCR were designed using the Universal Probe Library tool (Roche, Mannheim, Germany) and are listed (Table 2). Relative levels of target mRNAs were quantified using the comparative CT method and the 2^−ΔΔCT^ method [71,72], and normalized to the mRNA expression of β-Actin. Final mRNA levels were expressed as the normalized quantity of target transcript relative to the normalized quantity of the mRNA of the control group.

### 3.6. Immunohistochemical and Immunofluorescence Stainings

Human cryosections were allowed to air-dry for 1 h at RT. Human blood smears were let overnight to air-dry. The dried tissues and blood smears were later placed for fixation in 4% paraformaldehyde, or ice-cold methanol for 20 min, respectively. Afterwards they were washed three times with Phosphate-buffered saline (PBS) and three times with PBS supplemented with Tween 20 (PBST) to increase permeabilization. Mouse liver tissue sections from stored tissue in paraffin blocks were deparaffinized and rehydrated with xylene and decreasing graded ethanol, whereas antigen retrieval was induced by heating the sections in a 10 mM sodium citrate buffer (pH = 6.0) in a microwave for 20 min. Sections were let at RT to cool down and then rehydrated three times in double distilled water (ddH2O), and washed three times in PBST. Subsequently, both human and mouse liver tissues or blood smears were blocked for non-specific binding with 10% fetal calf serum (FCS) made in PBST supplemented with 1% bovine serum albumin (BSA) for 1 h at RT. For regular and fluorescence immunohistochemistry, sections were incubated overnight with respective antibodies. All primary and secondary antibodies used are listed in Table 1.

For regular immunohistochemistry, antibodies used were the goat polyclonal LCN2 (AF1757 from R&D Systems, Wiesbaden, Germany) for the human liver sections and blood smears and the polyclonal rabbit PLIN5 (PA1-46215 from Thermo Fisher Scientific) for tissue only. Both antibodies were used in dilution 1:50 in blocking solution. For the mouse sections, the same PLIN5 antibody was used, while for LCN2 detection the goat polyclonal AF3508 (R&D Systems) was used in dilution 1:40. The tissues and blood smears were incubated with the primary antibodies at 4 °C overnight for the detection of LCN2 and PLIN5.

After the primary antibody incubation, the sections were washed with PBST and followed washing with ddH2O before blocking endogenous peroxidase with 3% hydrogen peroxide (from Carl Roth, Karlsruhe, Germany) for 10–15 min to minimize background staining. After two washes with ddH2O and two with PBST, the sections were incubated for 1 h at RT with horse peroxidase (HRP)-conjugated secondary antibodies in dilution of 1:300. Washing in PBST followed, and peroxidase substrate was added on sections to induce staining. As substrate, SigmaFast™ 3,3′-Diaminobenzidine tablets (Sigma-Aldrich, Taufkirchen, Germany) were diluted in ddH_2_O and applied to tissues until color developed. Later on, the sections were submerged into ddH_2_O and then washed in PBST before they were counterstained with hematoxylin for 10 min. Finally, the stained sections were dehydrated with increasing graded ethanol and xylene before being mounted with DPX resin mounting solution (Sigma-Aldrich).

For the double staining with immunofluorescence, the antibodies used for the human and mouse liver sections were the same LCN2 and PLIN5 antibodies used in the regular immunostaining in dilution 1:50 in blocking solution. Moreover, the mouse monoclonal AFP antibody (MIA1301 from Thermo Fisher Scientific) in dilution 1:200, the rabbit polyclonal MPO antibody (AB9535 from Abcam, Cambridge, UK) in dilution 1:100 and the CD90 rabbit polyclonal antibody (orb229832 from Biorbyt, Cambridge, UK) in dilution 1:80. The tissues were incubated with a combination of the primary antibodies directed against LCN2-PLIN5, LCN2-AFP, LCN2-MPO and LCN2-CD90 at 4 °C overnight.

After the primary antibody incubation, the sections were washed with PBST and incubated for 1 h at RT with secondary antibodies (Table 1) conjugated to ALEXA Fluor dyes (Life Technologies, Thermo Fisher Scientific) in 1:300 in antibody dilution buffer. The slides were washed with PBST and mounted with Fluoroshield mounting medium with DAPI (4′,6-diamidino-2-phenylindole) (Abcam) and analyzed. All microscopy images were taken using the Nikon Eclipse 80i microscope (Nikon, Tokyo, Japan), equipped with the NIS-Elements Vis software (version 3.22.01, Nikon, Amsterdam, The Netherlands).

### 3.7. Statistical Analysis

Student’s *t*-test was applied for the statistical analyses of expression data for comparison of individual groups. Results are expressed as the mean ± standard deviation (SD). Probability values of ≤0.05 were considered statistically significant. All statistical analyses were performed using Excel™ Analysis 2010, Microsoft, Munich, Germany).

## 4. Conclusions

Collectively, in this study we report for the first time the presence of PLIN5 in HCC. The expression of PLIN5 launches in the tumor formation within HCC livers making it a promising marker for studies focusing on oncogenic effects in liver tissue. Moreover, this study shows that LCN2 which is already known as a pleiotropic protein increases its expression in an intensive manner within HCC livers in a non-specific way. Its main source is the injured neoplastic AFP-positive hepatocyte, as well as the inflammatory infiltrates and the hepatic progenitor cells. The different cells, positive for AFP, MPO or CD90, are responsible for various processes and HCC-related mechanisms. Thus, the multiple sources of LCN2 in the tumoral liver suggests that LCN2 is indeed pleiotropic, possibly participating in multiple functions in the tumor microenvironment, such as damage response, immunity and differentiation. Whereas PLIN5 for the first time was shown to associate with HCC tumor in this study, the precise mechanism in which it is regulated during the onset or progression of HCC is still unknown. Future studies analyzing the relation of LCN2 with its co-expressed markers in HCC, as well as the specific expression of PLIN5 in liver tumor are urgently needed to shed light on their distinct roles in the diagnosis, prognosis and possible treatment of HCC.

## Figures and Tables

**Figure 1 cancers-11-00385-f001:**
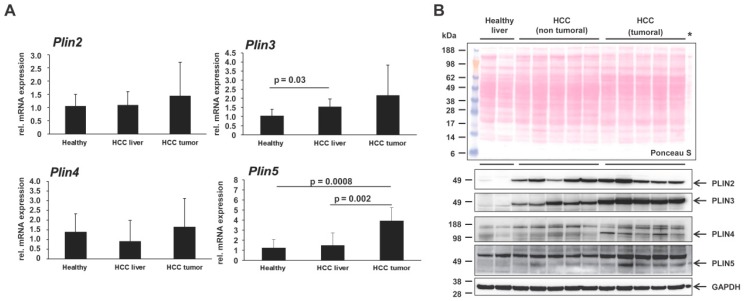
Expression of the perilipin family in liver and hepatocellular carcinoma (HCC). (**A**) Quantitative real-time PCR analysis of hepatic *Plin2*, *Plin3*, *Plin4* and *Plin5* expression in healthy mice (*n* = 4), non-tumoral liver (*n* = 8) and tumoral liver extracts (*n* = 8) of HCC mice. The quantity of mRNA in healthy mouse livers was set to 1, and expression levels in the HCC groups were expressed as relative values. All measurements were normalized to β-Actin expression. Primers are listed in Table 2. Statistical analysis was performed by Student t-test and standard deviations below 0.05 were considered significant. (**B**) Confirmation of differential expression of the perilipin proteins by Western blot analysis. Western blots were performed to detect PLIN2, PLIN3, PLIN4 and PLIN5 proteins in liver protein extracts from healthy mice (*n* = 3) and mice with HCC on tumoral and non-tumoral liver extracts (*n* = 8 animals/group). An extract of primary mouse hepatocytes (marked with asterisk), transient overexpressing PLIN5, was used to define the protein band of PLIN5 as it is lowly expressed in liver. Glycerinaldehyd-3-phosphate dehydrogenase (GAPDH) was used as a loading control. The antibodies used are listed in Table 1.

**Figure 2 cancers-11-00385-f002:**
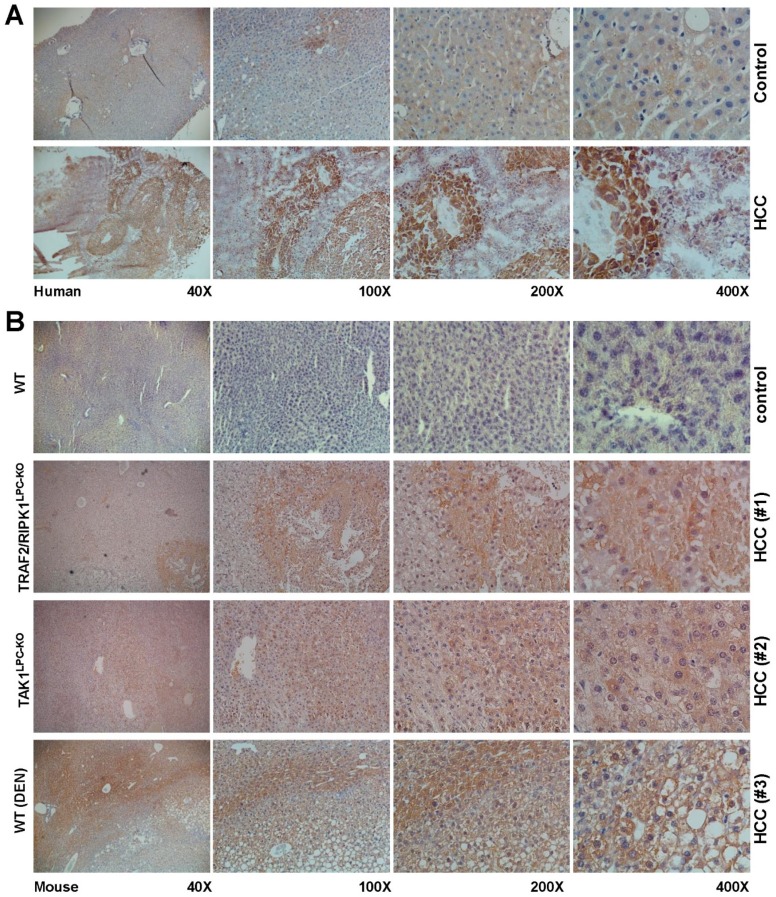
Immunohistochemical localization of PLIN5 in human livers of HCC patients and experimental mouse HCC models. Liver cryosections of (**A**) human hepatic biopsies and (**B**) paraffin sections of HCC mouse models were stained with an antibody against PLIN5. The depicted results are representative of all samples analyzed. As negative controls, either non-HCC human patient biopsies or hepatic sections from healthy mice were used. Magnifications presented are of 40×, 100×, 200× and 400×, respectively.

**Figure 3 cancers-11-00385-f003:**
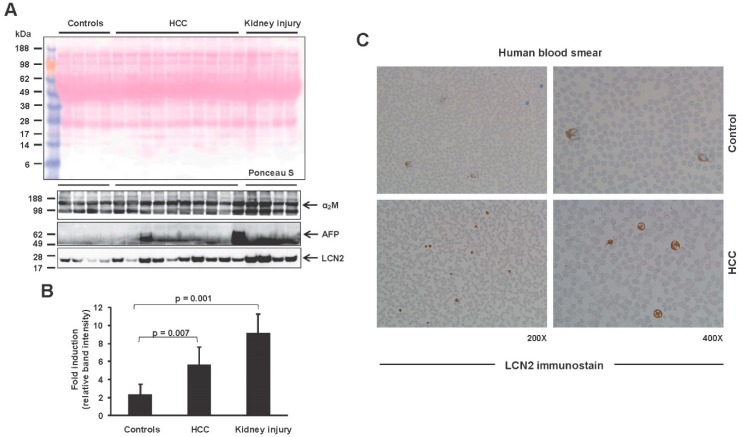
Lipocalin 2 (LCN2) expression in blood serum and blood smears withdrawn from HCC patients. (**A**) LCN2 protein secretion in the blood serum of human HCC patients. α-Fetoprotein (AFP) was shown as a known marker elevated in HCC patients, while α_2_-Macroglobulin (α_2_M) was used as a loading control. (**B**) Densitometry of LCN2 results depicted in (**A**). Statistical analysis was performed by Student t-test and standard deviations below 0.05 were considered significant. (**C**) Blood smears were stained for LCN2 by routine immunocytochemistry procedure to visualize LCN2 positive cells. Serum from healthy subjects was used as a control. Magnifications presented are 200× and 400×.

**Figure 4 cancers-11-00385-f004:**
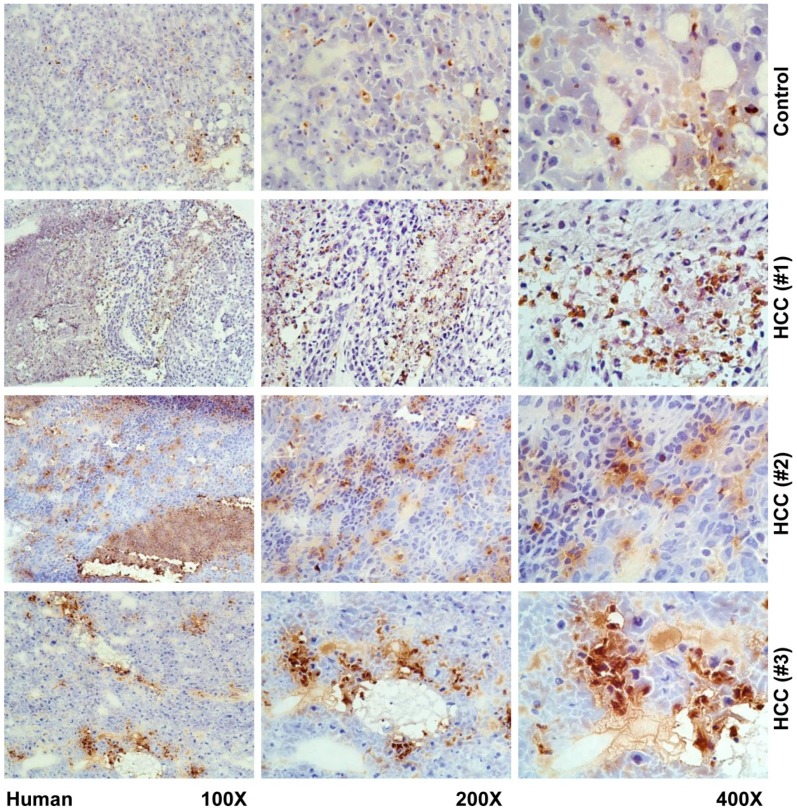
Immunohistochemical localization of LCN2 in human livers of HCC patients. Liver cryosections of human hepatic biopsies were stained with antibody against LCN2. Representative stains of three HCC patients and one control are depicted. As negative controls non-HCC human patient biopsies were used. Magnifications are 100×, 200× and 400×, respectively.

**Figure 5 cancers-11-00385-f005:**
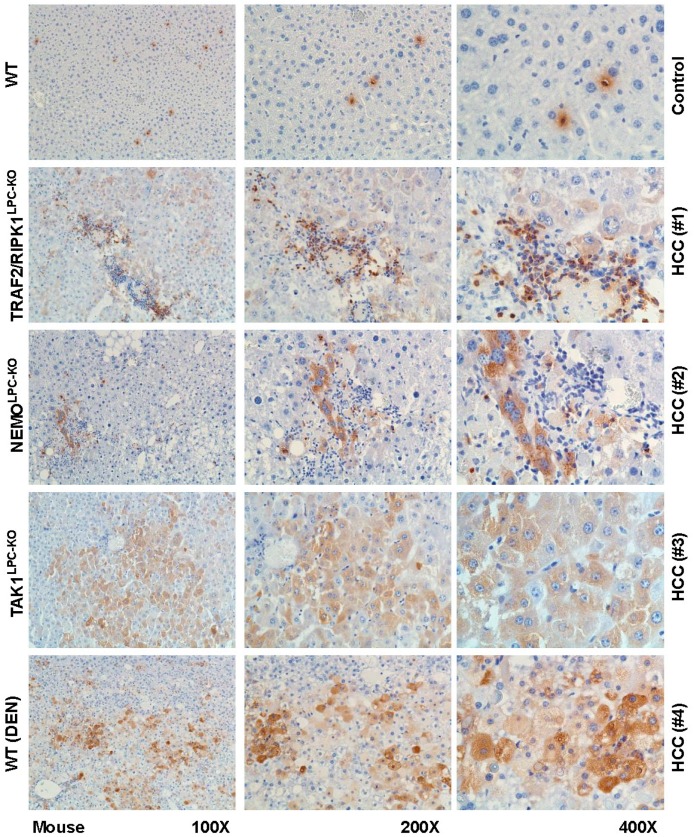
Immunohistochemical localization of LCN2 in livers of HCC mice. Liver paraffin sections of healthy and HCC mice were stained with antibody against LCN2. Representative images of one control and four HCC samples taken from different HCC models are depicted. Magnifications shown are 100×, 200× and 400×, respectively.

**Figure 6 cancers-11-00385-f006:**
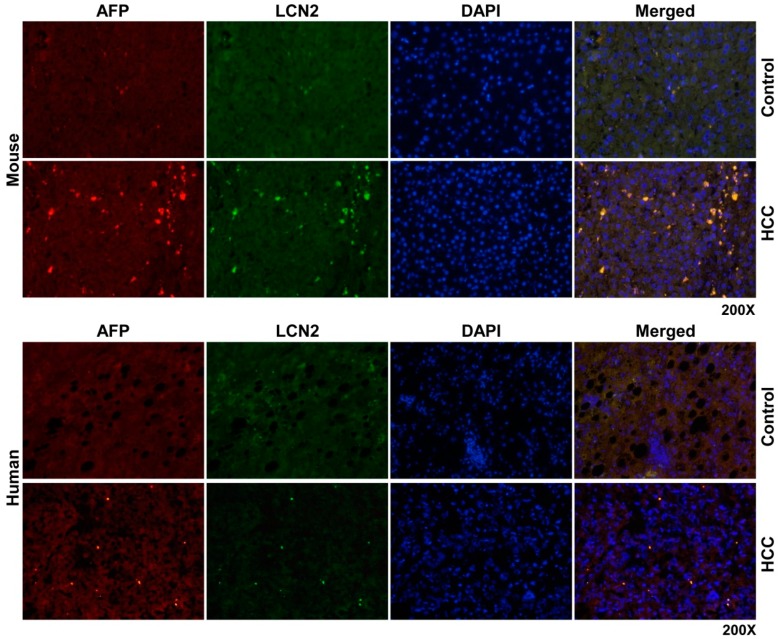
Double fluorescent Immunohistochemical staining of LCN2 and AFP in livers of HCC mice and human HCC liver biopsies. Liver paraffin sections of healthy and HCC mice as well as human liver biopsies of HCC or non-HCC patients were stained with antibodies against LCN2 and AFP. Alexa Fluor-conjugated secondary antibodies were used for visualization (green, LCN2; red, AFP). Nuclei were counterstained with DAPI (blue). Representative images are depicted. Magnification: 200×.

**Figure 7 cancers-11-00385-f007:**
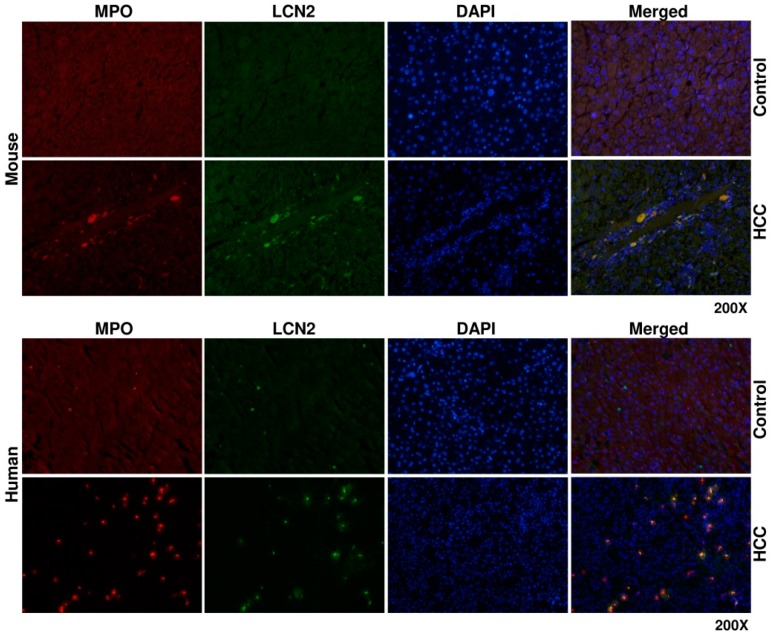
Double fluorescent immunohistochemical staining of LCN2 and MPO in livers of HCC mice and human HCC liver biopsies. Liver paraffin sections of healthy and HCC mice, as well as human liver biopsies of HCC or non-HCC patients were stained with antibodies against LCN2 and Myeloperoxidase (MPO). Alexa Fluor-conjugated secondary antibodies were used for visualization (green, LCN2; red, MPO) and nuclei were counterstained with DAPI (blue). Magnification: 200×.

**Figure 8 cancers-11-00385-f008:**
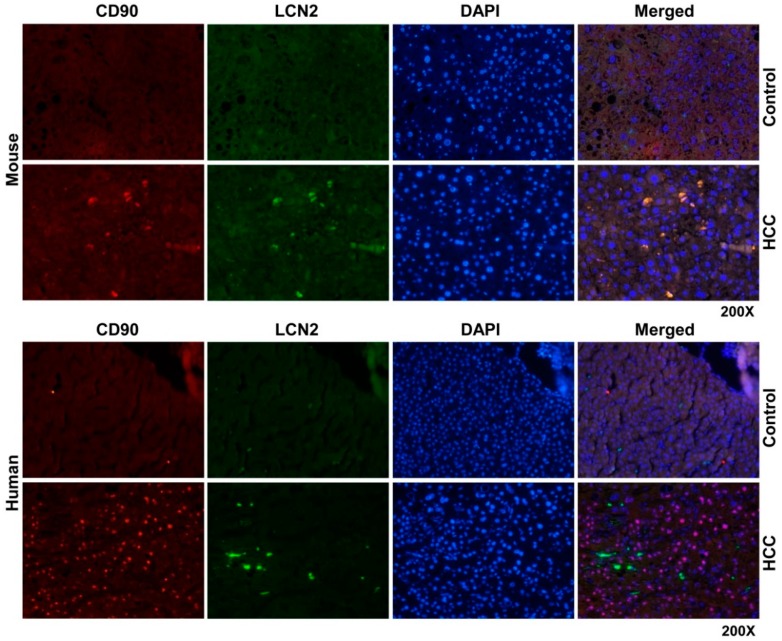
Double fluorescent immunohistochemical staining of LCN2 and CD90 in livers of HCC mice and human HCC liver biopsies. Liver paraffin sections of healthy and HCC mice as well as human liver biopsies of HCC or non-HCC patients were stained with antibodies against LCN2 and CD90. Alexa Fluor-conjugated secondary antibodies secondary antibodies were used for visualization (green, LCN2; red, CD90) and nuclei were counterstained with DAPI (blue). Magnification: 200×.

**Figure 9 cancers-11-00385-f009:**
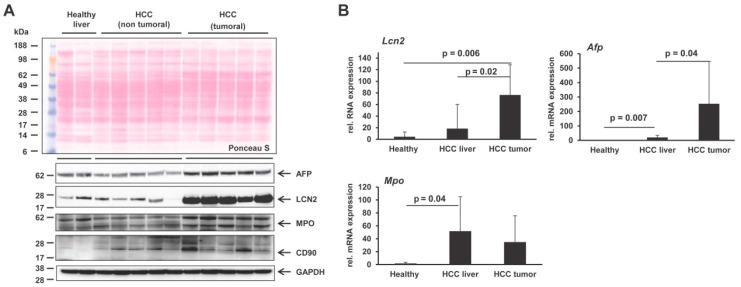
Differential expression of LCN2, AFP, MPO and CD90 in HCC mouse livers. (**A**) Western blots were performed to detect LCN2, AFP, MPO and CD90 in liver protein extracts from healthy mice (*n* = 3) and mice with HCC on tumoral and non-tumoral liver extracts (*n* = 8 animals/group). GAPDH was used as a loading control. The antibodies used are listed in Table 1. (**B**) Quantitative real-time PCR analysis of *Lcn2*, *Afp* and *Mpo* mRNA levels in healthy mice (*n* = 4), non-tumoral liver (*n* = 8), and tumoral liver extracts of HCC mice (*n* = 8). The quantity of mRNA in healthy mouse liver was set to 1, and expression levels in the HCC groups were expressed as relative values. All measurements were normalized to β-Actin expression. Primers are listed in Table 2. Statistical analysis was performed by the Student t-test and standard deviations below 0.05 were considered significant.

**Table 1 cancers-11-00385-t001:** Antibodies used for immunostaining and Western blot analysis.

Antibody	Cat. No.	Clonality */Host	Supplier	Species *
**Primary Antibodies**				
GAPDH	sc-32233	mono, mouse	Santa Cruz	h, m, rt
LCN2	AF3508	poly, goat	R&D Systems	m, rt
LCN2	AF1757	poly, goat	R&D Systems	h, m, rt
PLIN2	NB-110-40877	poly, rabbit	Novus Biologicals	h, m
PLIN3	NB-110-40765	poly, rabbit	Novus Biologicals	h, m
PLIN4	ABS526	poly, rabbit	Sigma Aldrich	m
PLIN5	PA1-46215	poly, rabbit	Thermo Fisher	h, m, b
MPO (Western blot)	HP9048	poly, rabbit	Hycult Biotech	h, m
MPO (Immunostaining)	AB9535	poly, rabbit	Abcam	h, m, rt, mk, p
CD90	orb229832	poly, rabbit	Biobyt	h, m, rt
AFP	MIA1301	mono, mouse	Thermo Fisher	h, m
α2-Macroglobulin	200-101-207-0100	poly, goat	Rockland Immunochemicals	h
**Secondary Antibodies**				
IgG-HRP	31460	poly, goat	Thermo Fisher	rb
IgG-HRP	sc-2005	poly, goat	Santa Cruz	m
IgG-HRP	31400	poly, mouse	Thermo Fisher	g
Alexa Fluor 488-IgG	A-11055	poly, donkey	Life Technologies	g
Alexa Fluor 555-IgG	A-21424	poly, goat	Thermo Fisher	m
Alexa Fluor 555-IgG	A-31572	poly, donkey	Life Technologies	rb

* Abbreviations used are: mono, monoclonal antibody; poly, polyclonal antibody; h, human; m, mouse; rt, rat; rb, rabbit; g, goat; p, pig; mk, monkey; b = bovine.

**Table 2 cancers-11-00385-t002:** Primers used for quantitative real time PCR.

Mouse Gene	Accession No.	Primer (5′→3′)
*β-Actin*	NM_007393	Forward: 5′-ctctagacttcgagcaggagatgg-3′
		Reverse: 5′-atgccacaggattccatacccaaga-3′
*LCN2*	NM_008491.1	Forward: 5′-ccatctatgagctacaagagaacaat-3′
		Reverse: 5′-tctgatccagtagcgacagc-3′
*MPO*	NM_010824.2	Forward: 5′-gatggaatggggagaagctc-3′
		Reverse: 5′-gcaggtagtcccggtatgtg-3′
*AFP*	NM_007423.4	Forward: 5′-gttctggcatgctgcaaa-3′
		Reverse: 5′-cctttgcaatggatgctctc-3′
*PLIN2*	M93275.1	Forward: 5′-ctccactccactgtccacct-3′
		Reverse: 5′-gcttatcctgagcaccctga-3′
*PLIN3*	NM_025836.3	Forward: 5′-ccacaggatgctgaaaagg-3′
		Reverse: 5′-tgatgtccctgaacatgctg-3′
*PLIN4*	NM_020568.3	Forward: 5′-ggacttacaaacagcaacagacc-3′
		Reverse: 5′-tctgtgagttggtggacacttt-3′
*PLIN5*	NM_025874.3	Forward: 5′-gtcggagaagctggtggac-3′
		Reverse: 5′-tcagctgccaggactgcta-3′

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
