# Peer review of "Perilipin 5 and Lipocalin 2 Expression in Hepatocellular Carcinoma"

_cancers, 2019, doi:10.3390/cancers11030385_

Round 1
Reviewer 1 Report
Perilipin-5 and Lipocalin 2 expression in hepatocellular carcinoma
Cancers- MDPI
Article
ID: cancers-439715
I read with great interest the above titled article. However, there are several issues that need to be addressed by the authors.
1. Abstract: second line (first page, line 13): the statement “with active research on pathogenesis and treatment” is a sudden statement and fitting with the early part of the sentence. The whole manuscript needs editing for academic English and grammatical errors.
2. Abstract: Line 17: Which “models”?
3. Abstract: Line 20: The statement, “including suppression of invasiveness and metastasis” not clearly written.
4. Abstract: Line 26: “Therapeutic targets” or in the follow up and detection?
5. Introduction: Page 1, line 38: should be “first studies pointed to …” not point to. Also if there are several studies you may need to add another reference to ref number 7.
6. Page 2: Line 46, grammatical errors “to achieve that [9]”.
7. Page 2: Line 83: “AFP-positive hepatocytes represent the main source for the massive expression of LCN2 in tumor tissue.” Do we know other sources/causes for expression of LCN2 in these tumor cells?
8. Page 9: Lines 195-197: The statement, “Since AFP……” is this your hypothesis, you might need to strengthen your argument for such assumption by adding supportive elements.
9. Page 11: Line 246: add the reference for these guidelines or the link as a reference.
10. Page 11: line 253: state the causes for cirrhosis as they are were Hepatitis B and Hepatitis C negative.
Author Response
see pdf file

Reviewer 2 Report
The authors studied perilipin-5 and lipocalin-2 expression in HCC. The results of patients based on very limited patient numbers. The study design was weak. No detailed mechanisms were provided. Most results are based on simple observation without statistical analyses. The weak data certainly could not lead to the conclusion the authors tried to make (the strong presence of PLIN5 and LCN2 in HCC …… could establish them as therapeutic targets against HCC).
Key comments:
1. On page 3, the authors stated “the Plin5 was strongly and significantly upregulated in HCC liver”. However, fig. 1A only showed the increase in expression in the HCC tumor, no in the normal liver of mice with HCC.
2. On pages 5-7, the authors interpreted results of figures 3-5 without any statistical analyses. Please provide statistical analysis and then interpret the results accordingly.
3. For results of patients, the case numbers are generally too low to draw any conclusions.
4. In Fig. 6, control patients seemed to have higher LCN2 expression compared to HCC patients. This finding contradicted the previous findings. In addition, the photos of patients’ samples are not convincing about co-staining of AFP and LCN2 at the same cells.
Author Response
see pdf file

Reviewer 3 Report
Abnormal lipometabolism occurs in various tumors and generates a series of changes in these molecules. Therefore, different kinds of pathological changes are observed. PLIN1, PLIN2, and PLIN3 are co-expressed in HCC
In this study, the authors show for the first time that PLIN5 is strongly expressed in tumors of human patients with HCC as well as in mouse livers of experimental HCC models. They have also already show that hepatic PLIN 5 expression could be regulated by LCN2 (Asimakopoulou A et al Biochim Biophys Acta 2014). The presence of this protein in blood and urine, in combination with the presence of α-Fetoprotein (AFP), is hypothesized to serve as a biomarker of early stages of HCC. In the current study, they show in humans and mice that LCN2 is secreted into the serum from liver cancer tissue. They also show that AFP-positive hepatocytes represent the main source for the massive expression of LCN2 in tumoral tissue.
Minor revision: The authors don’t really show that these biomarker are associated with early stages of HCC. Only 9 human liver tissues sections were analysed. The authors should moderate the conclusions (“early stage and tumor progression”). They should also confirm the result on a large cohort with I possible several samples: normal liver, fatty liver, cirrhotic non tumoral liver, HCC at different stages (BCLC A, B or C) and if possible pre neoplasic nodule.
The authors don’t really show a co expression of PLIN5 and LCN2 in the same AFP positive HCC human or murine cell. So they should also moderate the conclusions.
I’m agree with the sentence that “the strong presence of PLIN5 and LCN2 in HCC and understanding their roles could establish them as potential biomarker and/or therapeutic targets against HCC.
Author Response
see pdf file

Round 2
Reviewer 2 Report
The authors have adequately revised the manuscript.